# Acute Effects of Lixisenatide on Energy Intake in Healthy Subjects and Patients with Type 2 Diabetes: Relationship to Gastric Emptying and Intragastric Distribution

**DOI:** 10.3390/nu12071962

**Published:** 2020-07-01

**Authors:** Ryan Jalleh, Hung Pham, Chinmay S. Marathe, Tongzhi Wu, Madeline D. Buttfield, Seva Hatzinikolas, Charles H. Malbert, Rachael S. Rigda, Kylie Lange, Laurence G. Trahair, Christine Feinle-Bisset, Christopher K. Rayner, Michael Horowitz, Karen L. Jones

**Affiliations:** 1Endocrine and Metabolic Unit, Royal Adelaide Hospital, Adelaide SA 5000, Australia; ryan.jalleh@sa.gov.au (R.J.); chinmay.marathe@adelaide.edu.au (C.S.M.); tongzhi.wu@adelaide.edu.au (T.W.); michael.horowitz@adelaide.edu.au (M.H.); 2Adelaide Medical School, Centre of Research Excellence in Translating Nutritional Science to Good Health, The University of Adelaide, Adelaide SA 5000, Australia; hung.pham@adelaide.edu.au (H.P.); seva.hatzinikolas@adelaide.edu.au (S.H.); rachael.tippett@adelaide.edu.au (R.S.R.); kylie.lange@adelaide.edu.au (K.L.); laurence.trahair@adelaide.edu.au (L.G.T.); christine.feinle@adelaide.edu.au (C.F.-B.); chris.rayner@adelaide.edu.au (C.K.R.); 3School of Health Sciences, University of South Australia, Adelaide SA 5001, Australia; madeline.buttfield@bensonradiology.com.au; 4Aniscan, Institut National de la Rechercher Agronomique, 35590 Saint-Gilles, France; charles-henri.malbert@inra.fr; 5Department of Gastroenterology and Hepatology, Royal Adelaide Hospital, Adelaide SA 5000, Australia

**Keywords:** lixisenatide, intragastric meal retention, energy intake, type 2 diabetes

## Abstract

Glucagon-like peptide-1 receptor agonists induce weight loss, which has been suggested to relate to the slowing of gastric emptying (GE). In health, energy intake (EI) is more strongly related to the content of the distal, than the total, stomach. We evaluated the effects of lixisenatide on GE, intragastric distribution, and subsequent EI in 15 healthy participants and 15 patients with type 2 diabetes (T2D). Participants ingested a 75-g glucose drink on two separate occasions, 30 min after lixisenatide (10 mcg) or placebo subcutaneously, in a randomised, double-blind, crossover design. GE and intragastric distribution were measured for 180 min followed by a buffet-style meal, where EI was quantified. Relationships of EI with total, proximal, and distal stomach content were assessed. In both groups, lixisenatide slowed GE markedly, with increased retention in both the proximal (*p* < 0.001) and distal (*p* < 0.001) stomach and decreased EI (*p* < 0.001). EI was not related to the content of the total or proximal stomach but inversely related to the distal stomach at 180 min in health on placebo (*r* = −0.58, *p =* 0.03) but not in T2D nor after lixisenatide in either group. In healthy and T2D participants, the reduction in EI by lixisenatide is unrelated to changes in GE/intragastric distribution, consistent with a centrally mediated effect.

## 1. Introduction

Glucagon-like peptide-1 (GLP-1) receptor agonists (RAs)—both ‘short’ and ‘long’ acting—induce moderate weight loss in obese subjects with or without type 2 diabetes (T2D) [1,2,3,4,5]. The mechanisms underlying this weight loss are poorly defined. ‘Short-acting’ GLP-1RAs slow gastric emptying markedly [6,7], which appears to be primarily responsible for their effect to diminish postprandial glycaemia substantially [8]. ‘Long-acting’ GLP-1RAs, which are used widely in the management of obesity, probably have a lesser effect on gastric emptying, so that both preprandial and postprandial glucose lowering may be attributable primarily to their insulinotropic and glucagonostatic properties [9]; however, it is now clear that long-acting GLP-1RAs do slow gastric emptying [10]. GLP-1, secreted from L-cells in the epithelium of the small intestine, binds to GLP-1 receptors that are expressed in multiple organs, including the pancreatic islets, kidneys, lungs, heart, and central and peripheral nervous systems [11]. Circulating GLP-1 is able to access central GLP-1 receptors in areas not fully blocked by the blood–brain barrier, such as the subfornical organ and area postrema [12,13]. Most GLP-1RAs are able to activate central GLP-1 receptors expressing neurons. Larger molecule GLP-1RAs that are unable to cross the blood–brain barrier appear to act via secondary signals from the vagus nerve [14], while smaller molecule GLP-1RAs, including lixisenatide, pass through the blood–brain barrier directly [15]. Possible mechanisms underlying weight loss by GLP-1RAs include satiation induced by the slowing of gastric emptying and a consequent prolongation of gastric distension, centrally-mediated anorexia, and the induction of nausea as an adverse effect [16]. The stomach comprises distinct anatomical regions—the fundus, body (corpus), antrum and pylorus, with the proximal stomach incorporating the fundus and proximal corpus and the distal stomach incorporating the distal corpus and antrum. The proximal stomach is primarily responsible for the storage of food and relaxes in response to eating to accommodate a meal with only a modest change in intragastric pressure, whereas the antrum grinds solid food into small particles, usually <1 mm in size, that are delivered into the small intestine at a rate that optimises digestion and absorption. Accordingly, each region plays a coordinated role in the regulation of gastric emptying. In health, antral—rather than proximal or total—intragastric content is most closely related to energy-intake suppression in young and older subjects, probably indicative of an effect of antral distension [17]. The effects of GLP-1RAs on intragastric meal distribution and the relationship of changes in energy intake with gastric emptying/intragastric distribution have not been evaluated.

The aims of this study were to evaluate the acute effects of lixisenatide on intragastric distribution and subsequent energy intake in health and T2D. This was a prespecified secondary analysis from a study evaluating the effects of lixisenatide on gastric emptying and blood pressure in these groups [8].

## 2. Materials and Methods 

Twenty-four ‘healthy’ participants and 74 participants with T2D, managed by diet or a stable dose of metformin alone, were ‘prescreened’ by phone or email interview. Participants were required to be 40–80 years of age, with BMI 19–35 kg/m^2^ and, for T2D patients, have an HbA1c < 8.5% (<69 mmol/mol). Three healthy participants and 47 T2D participants were excluded. Full exclusion criteria have been published [8]. The remaining participants attended the Royal Adelaide Hospital (RAH) for a screening visit and had a venous blood sample taken for measurement of HbA1c, liver function, creatinine, glucose, and biochemistry and, for females, a urine test for pregnancy. Of the 21 ‘healthy’ participants, 18 were enrolled and 3 were excluded; of the 27 participants with T2D, 16 were enrolled. Of the healthy participants, 2 withdrew due to adverse events (nausea soon after administration of lixisenatide) and 1 was withdrawn on the first study day because of a low baseline blood pressure. Of the T2D participants, 1 was withdrawn due to inability to attend the RAH on the two study days.

Hence, a total of 15 healthy participants (9 male, 6 female; age: 67.2 ± 2.3 years; body mass index: 25.4 ± 0.8 kg/m^2^) and 15 participants with T2D managed by diet or metformin alone (9 male, 6 female; age: 61.9 ± 2.3 years; BMI: 30.3 ± 0.7 kg/m^2^; duration of known diabetes: 5.3 ± 1.2 years; HbA_1_c: 6.9 ± 0.2% (51.8 ± 2.3 mmol/mol) were studied. Ten of the 15 participants in the T2D group were taking metformin (plasma half-life: 4–9 h [18]) that was withheld for 48 h prior to the study because of its potential effect on gastric emptying [19]. The other 5 participants were managed by diet alone. Antihypertensive medication (used by 3 age-matched, non-diabetic controls and 4 participants with T2D) was also held for 48 h. All participants were nonsmokers, and none had a history of gastrointestinal disease or surgery, significant respiratory, cardiac, hepatic and/or renal disease, alcohol consumption >20 g per day or epilepsy, and none were unable to withhold any medication likely to influence blood pressure or gastrointestinal function. The sample size was based on a primary outcome presented in a previous published study [8] and this is a secondary analysis, as stated previously.

### 2.1. Protocol

The study followed a randomised, double-blind, placebo-controlled, crossover design. Using a stratified (healthy/T2D), randomised permuted-blocks (block size of 2) method, participants were allocated to their respective groups by Sanofi. Participants attended the Department of Nuclear Medicine, Positron Emission Tomography and Bone Densitometry at the RAH at 8.30 am after an overnight fast (14 h for solids, 12 h for liquids) on two separate occasions.

Participants received either lixisenatide (10 mcg) or placebo subcutaneously (sc), and 27 min later ingested a drink comprising 75 g glucose (280.5 kcal) radiolabelled with 20 MBq ^99m^Tc-Calcium Phytate (Radpharm Scientific, Belconnen, ACT, Australia), made up to 300 mL water, within 3 min (*t* = 0 min was defined as the end of drink ingestion).

### 2.2. Measurements

Gastric emptying was measured by scintigraphy for 180 min. Data was acquired every minute for the first hour, then every 3 min for the subsequent 2 h. A region-of-interest was drawn around the total stomach, which was then divided into proximal and distal stomach regions to determine intragastric distribution, i.e., retention of the drink in the proximal and distal stomach regions, whereby the proximal region corresponded to the fundus and proximal corpus and the distal stomach region corresponded to the antrum and distal corpus [20]. Data were analysed, using purpose-built software (CH Malbert, LabView, National Instruments (NI), Dallas TX, USA, 2013), by two experienced nuclear medicine technologists (MDB, KLJ), blinded to the study conditions.

Energy intake was assessed from *t* = 180 min when each participant was offered a cold, buffet-style meal on a tray and allowed to eat for 30 min until they felt comfortably full [21]. The buffet meal comprised four slices (125 g) of wholemeal bread, four slices (125 g) of white bread, 100 g sliced ham, 100 g sliced chicken, 85 g sliced cheddar cheese, 100 g lettuce, 100 g sliced tomato, 100 g sliced cucumber, 20 g mayonnaise, 20 g margarine, 170 g apple, 190 g banana, 200 g strawberry yogurt, 150 g chocolate custard, 140 g fruit salad, 600 mL iced coffee, 500 mL orange juice and 600 mL water with a total energy content of 11 808 kJ. Food was weighed prior to consumption and the amount (kcal) of energy consumed was derived using commercial software (Foodworks 3.01, Xyris Software, Highgate Hill, QLD, Australia), based on the weight of the remaining food on the tray at the end of the 30-min period [21].

Nausea was assessed using a validated 100 mm visual analog questionnaire [22], prior to study-drug administration, before consumption of the glucose drink and at 15-min intervals during the gastric emptying measurement. These data, together with plasma glucose, insulin, C-peptide, and glucagon concentrations, have previously been reported [8].

The protocol was approved by the Human Research Ethics Committee of the Royal Adelaide Hospital, and each participant provided written, informed consent. All studies were carried out in accordance with the Declaration of Helsinki. The study was registered on clinicaltrials.gov (NCT: 02308254).

### 2.3. Statistics

Effects of treatment and group were assessed with two-way repeated measures analysis of variance (ANOVA), with treatment as a within-subject factor and group as a between-subject factor, including treatment and group main effects and the treatment by group interaction. Relationships between energy intake and the content of the total, proximal and distal stomach after placebo and lixisenatide were assessed using linear regression analysis. Data were analysed using SPSS Statistics (SPSS, Chicago, IL, USA) and are presented as means ± SEMs. A value of *p* < 0.05 was considered significant.

## 3. Results

The studies were well tolerated. As reported, scores for nausea were uniformly very low with no difference between placebo and lixisenatide, and lixisenatide slowed gastric emptying markedly in both groups (*p* < 0.001). The proximal stomach retention at 180 min in the healthy group was 6.6 ± 3.5% with placebo vs. 40.9 ± 4.6% with lixisenatide (*p* < 0.001) and in the T2D group was 6.3 ± 4.1% with placebo vs. 34.8 ± 24.5% with lixisenatide (*p* < 0.001). The distal stomach retention at 180 min in health was 9.4 ± 9.1% with placebo vs. 18.6 ± 11.1% with lixisenatide (*p* < 0.001) and in T2D was 8.2 ± 4.2% with placebo vs. 19.6 ± 10.4% with lixisenatide (*p* < 0.001) (Figure 1). There was no difference in the effect of lixisenatide on intragastric distribution between the two groups.

Lixisenatide decreased energy intake (*p* < 0.001) in both healthy participants and T2D (Figure 2) by −29.2 ± 4.0% and −27.1 ± 8.2%, respectively. On the placebo day, there was no relationship between energy intake and the content of the proximal (*r* = 0.005, *p* = 0.99 in healthy participants; *r* = 0.11, *p* = 0.70 in T2D) or total stomach (*r* = −0.47, *p* = 0.09 in healthy participants; *r* = −0.11, *p* = 0.70 in T2D) at *t* = 180 min after placebo. However, energy intake was inversely related to the distal stomach content at *t* = 180 min in health (*r* = −0.58, *p* = 0.03) but not in T2D (*r* = −0.31, *p* = 0.27). On the lixisenatide day, there was no relationship between energy intake and the distal stomach content in healthy participants (*r* = −0.16, *p* = 0.58) or T2D (*r* = −0.004, *p* = 0.99) (Figure 3). Similarly, there was no relationship between energy intake and the proximal stomach content (*r* = −0.16, *p* = 0.58 in healthy participants; *r* = 0.10, *p* = 0.71 in T2D) or total stomach content (*r* = −0.23, *p* = 0.42 in healthy participants; *r* = 0.09, *p* = 0.75 in T2D).

## 4. Discussion

Our study evaluated the acute effects of the ‘short-acting’ GLP-1RA lixisenatide on energy intake at a buffet meal, intragastric distribution of a glucose drink, and the relationship between them. We used a lower dose of lixisenatide (10 mcg) compared to the dose used clinically as monotherapy (titrated gradually to 20 mcg) to maximise tolerability. Doses of lixisenatide less than 20 mcg are, however, frequently used in practice, particularly in combination with insulin glargine, which has been shown to be well tolerated and associated with weight loss [23,24].

We have reported that lixisenatide at a dose of 10 mcg slows gastric emptying markedly in both health and well-controlled T2D, associated with a reduction in glycaemia [8]. The current study establishes that lixisenatide also affects intragastric distribution by increasing retention in both the proximal and distal stomach and reduces energy intake at a subsequent buffet meal. However, the effect of lixisenatide on energy intake was unrelated to its profound effects on intragastric distribution or total stomach emptying, strongly supporting the concept that the observed reduction of energy intake is primarily centrally mediated. We confirmed that, in health, the suppression of energy intake following a nutrient preload is closely related to the content of the distal—but not the total or proximal—stomach, presumably indicative of antral distension being a key determinant [17].

Intracerebroventricular injection of GLP-1 inhibits feeding in fasted rats [25], and radioligand binding studies in rats have shown high densities of GLP-1 receptors in the brain, including areas thought to be responsible for satiation [26]. In humans, cells positive for GLP-1 mRNA are expressed widely in the brain, including the hypothalamus, which is pivotal to the regulation of appetite [27]. One study, using functional MRI, has reported diminished responses in appetite- and reward-related brain areas after administration of intravenous exenatide in normoglycemic obese and T2D subjects, correlating with a reduction in food intake [28]. It has also been hypothesised that GLP-1 has peripherally mediated effects on appetite [16], particularly via slowing of gastric emptying, with consequent activation of gastric mechano-receptors which relay action potentials via the vagal nerves to the nucleus of the solitary tract to suppress appetite [29]. Our study does not support this hypothesis, given the absence of a relationship between energy intake and the increased retention of gastric content after lixisenatide. It has been suggested that weight loss resulting from GLP-1RAs represents an adverse effect due to the induction of nausea [30,31]. This was clearly not the case in our study, where there was minimal nausea after treatment with either lixisenatide or placebo.

Our study has several strengths, including the randomised, double-blind, placebo controlled, cross-over design in both health and T2D and use of the ‘gold standard’ technique of scintigraphy to measure gastric emptying and intragastric distribution.

Limitations relate to the ‘proof-of-concept’ design, the use of a glucose drink rather than a more physiologic mixed solid/liquid meal and that only the effects of a single, low dose of lixisenatide—instead of sustained administration—were assessed. Evidence to suggest that the suppression of energy intake by lixisenatide is centrally mediated is also indirect, and other mechanisms that could contribute to a reduction in energy intake by lixisenatide, including stimulation of brown adipose tissue activity, were not evaluated [32,33]. It should also be appreciated that the failure to observe a significant relationship between energy intake and the distal stomach content in the T2D group on the placebo days may reflect the modest number of participants, particularly given the relative heterogeneity of this group.

## 5. Conclusions

In conclusion, acute administration of 10 mcg lixisenatide reduces energy intake in the absence of nausea, slows gastric emptying of a glucose drink, and increases retention in the distal and proximal stomach. The reduction in energy intake by lixisenatide was unrelated to changes in gastric emptying/intragastric distribution, consistent with a centrally mediated effect.

## Figures and Tables

**Figure 1 nutrients-12-01962-f001:**
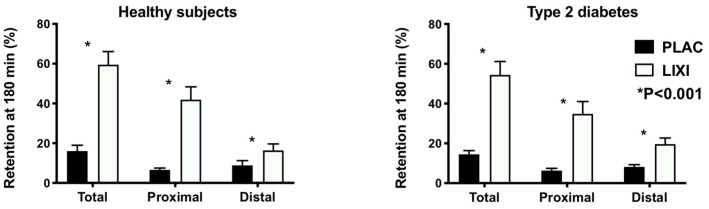
Intragastric distribution of gastric content (retention in the total, proximal, and distal stomach regions) at *t* = 180 min following a 75-g glucose drink radiolabelled with 20 MBq ^99m^Tc-Calcium Phytate in health and type 2 diabetes (T2D) following lixisenatide (10 mcg sc) or placebo (sc). *p* < 0.001 treatment difference in two-way repeated measures ANOVA. Treatment-by-group interactions all nonsignificant (*p* > 0.05).

**Figure 2 nutrients-12-01962-f002:**
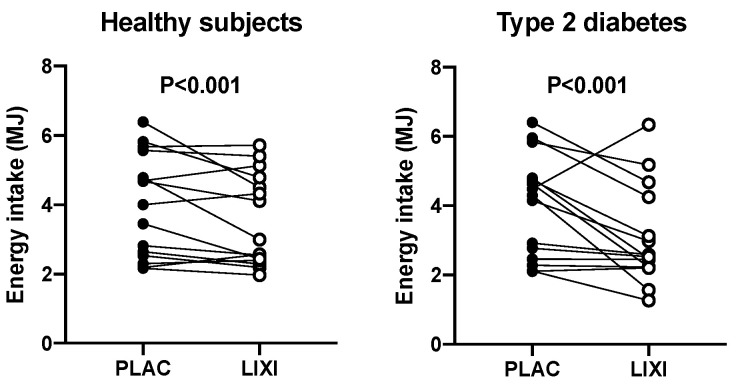
Effect of lixisenatide (10 mcg sc) (open circles) vs. placebo (sc) (black circles) on energy intake (MJ) at a buffet-style meal in healthy participants and patients with T2D *p* < 0.001 for both (placebo vs. lixisenatide). *p* < 0.001 treatment difference in two-way repeated measures analysis of variance (ANOVA). Treatment-by-group interaction nonsignificant (*p* > 0.05).

**Figure 3 nutrients-12-01962-f003:**
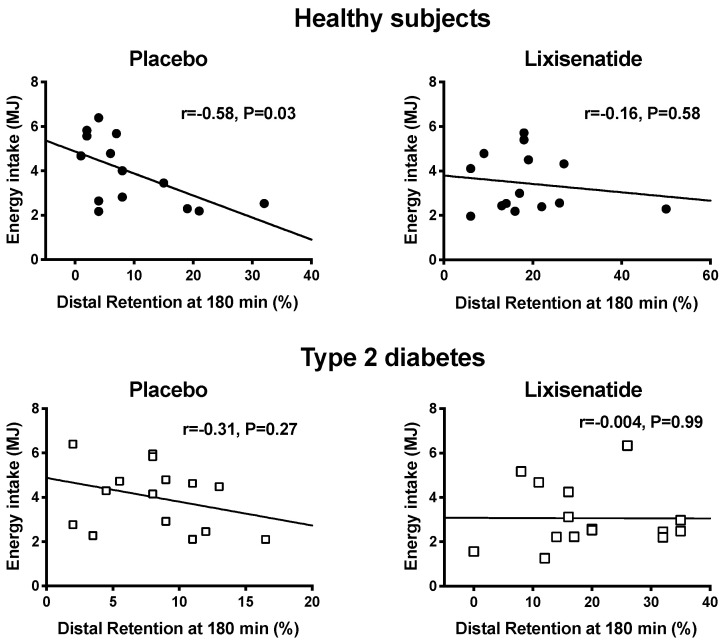
Relationships between energy intake (MJ) consumed at the buffet-style meal and retention in the distal stomach at 180 min after a drink containing 75 g glucose in healthy participants (black circles) following placebo (*r* = −0.58, *p* = 0.03) and lixisenatide (*r* = −0.16, *p* =0.58) and patients with T2D (open squares) following placebo (*r* = −0.31, *p* = 0.27) and lixisenatide (*r* = 0.004, *p* = 0.99).

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
