# Peer review of "Acute Effects of Lixisenatide on Energy Intake in Healthy Subjects and Patients with Type 2 Diabetes: Relationship to Gastric Emptying and Intragastric Distribution"

_nutrients, 2020, doi:10.3390/nu12071962_

Round 1

Reviewer 1 Report

Introduction

Page 2 Line 47 – However, it is…. – sentence is unclear and should be reworded and/or shortened.

Methods

Was any exclusion criteria applied during recruitment process?

Method of calculating energy intake is not adequately described, please provide details. 

What was the composition of the buffet meal e.g. types of nutrients on offer?

Was assessment of gastric emptying done in blinded manner by more than one assessor? Dividing stomach regions of patients can be variable and subjective.

How was intragastric distribution of food correlated with ‘retention’? What is the definition of ‘retention’? How was intragastric distribution calculated for each region? The methods description is not provided (even briefly).

Results

Statistical significance is not shown on Figure 1 graphs.  

Statistical significance shown on Figure 2 suggests paired analysis. Was this the case? If so, what was the post-hoc analysis used? Authors should also show individual values and pairings in separate graph. What was the percentage decrease (between placebo and lix) in each group (healthy vs. T2D)?

Figure legends are not descriptive enough and require further details.  

Author Response

Introduction:

  1. The sentence has now been reworded (pg 2, lines 50-51).

Methods:

  1. The data presented in this paper represent a secondary outcome as discussed pg 2, lines 73-75. Full details of the exclusion criteria applied during the recruitment process are described fully in Jones KL et al. Effects of lixisenatide on postprandial blood pressure, gastric emptying and glycaemia in health and type 2 diabetes. Diabetes Obes Metab 2019;21:1158-67. We have stated this on pg 2, lines 80-81.
  2. The composition of the buffet meal and the method of calculating energy intake have now been incorporated into the manuscript (pg3, lines 120-128).
  3. Assessment of gastric emptying and intragastric distribution was performed by one investigator (MDMB) and subsequently checked by another (KLJ) i.e. all data were analysed by two investigators (both trained in nuclear medicine). Both investigators were blinded to the study condition (pg 3, lines 116-118).
  4. Intragastric distribution refers to ‘retention’ in the proximal and distal stomach regions i.e. how the food within the total stomach has been distributed. This is now described in the manuscript (pg 3, lines 114-116).

Results:

  1. Figure 1 has been amended to include statistical significance (pg 4).
  2. Figure 2: The statistical analysis was performed by a professional biostatistician who is a co-author on the manuscript (KL). The P value relates to a significant treatment effect from the two-way ANOVAs. No post-hoc analyses were required as the treatment by group interaction was not significant for all three analyses. Figure 2 (pg 4) has been replaced with graphs indicating individual values and pairings. The percentage of decrease has also been included in the text (pg 4, line 155).
  3. The figure legends have been amended to include more detail as requested (pg 4 and 5).

Reviewer 2 Report

Jalleh and co-authors are presenting results on energy intake, gastric emptying and intragastric distribution in healthy individuals and in patients with type 2 diabetes. These outcomes were prespecified as secondary outcomes. The study is well-designed, and the use of scintigraphy to assess gastric emptying is a strength. The manuscript is nicely and clearly written, the data are well presented and discussed.

General comments:

It would be helpful for the reader if the physiological relevance of proximal vs. distal stomach content/retention was explained in the introduction.

Also, it would be relevant in the methods section to describe how gastric emptying was determined, alternatively provide a reference to how it was done. Was the analysis done automatically or manually, and if manually, by one or two researchers. A bit more detail would be beneficial.

Regarding the results (Figure 1 and 2), the authors could be consistent about providing p-values in the panels.

Regarding withdrawal of Metformin for 48 hours, it would be relevant to state why 48 hours was chosen, maybe mention the half-life of Metformin and provide a reference that 48 hours is enough time to avoid any influence of the drug on the outcomes.

Minor comment:

The sentence on line 47 starting with “However” is unclear and should be rephrased.

Author Response

We thank the reviewer for their constructive criticisms.

In response to the issues raised by the reviewer:

General comments:

  1. More information relating to the physiology of the proximal and distal stomach has now been included in the Introduction (pg 2, line 61-68).
  2. The methodology used to assess gastric emptying and intragastric distribution has been expanded on in the manuscript. All of the data were assessed by an individual investigator and subsequently checked by another highly experienced investigator, both blinded to the experimental conditions (pg 3, lines 116-118).
  3. P values have now been provided for Figure 1 (pg 4).
  4. Metformin was withdrawn for 48 hours ie ~ 4 half lives due to potential effects on gastric emptying. This has now been included in the manuscript (pg 2, lines 91-93) and a two references provided (ref 18 and 19).

Minor comment

  1. The sentence has now been reworded (pg 2, lines 50-51).